# On the regulation of arterial blood pressure by an intracranial baroreceptor mechanism

Philippa Wittenberg[1], Fiona D. McBryde[2], Alla Korsak[1], Karla L. Rodrigues[1] , Julian F. R. Paton[2], Nephtali Marina[1,3] and Alexander V. Gourine[1]

[1]*Centre for Cardiovascular and Metabolic Neuroscience, Neuroscience, Physiology & Pharmacology, University College London, London, UK*
[2]*Manaaki Manawa Centre for Heart Research, Department of Physiology, Faculty of Medical and Health Sciences, University of Auckland, Auckland, New Zealand*
[3]*Division of Medicine, University College London, London, UK*

Handling Editors: Kim Barrett & Philip Ainslie

The peer review history is available in the Supporting Information section of this article (https://doi.org/10.1113/JP285082#support-information-section).

*The Journal of Physiology*

**Abstract figure legend** This experimental animal (rat) study describes the regulation of systemic arterial blood pressure by an intracranial baroreceptor mechanism sensitive to changes in brain perfusion. The animals were instrumented for physiological monitoring with access to the brain ventricular system to record intracranial pressure (ICP) and experimentally manipulate ICP (within the physiological range of 0–30 mmHg). Activation of the intracranial baroreceptor mechanism by raised ICP triggers robust, non-habituating, proportional increases in sympathetic nerve activity and mean arterial blood pressure (MAP), which maintain cerebral perfusion pressure (CPP) and brain blood flow. The study shows a linear relationship between MAP and ICP in both anaesthetized and conscious rats and describes the effect of intracranial baroreceptor activation on the arterial baroreflex.

**Abstract**   Maintaining sufficient cerebral blood flow is critical for brain function. There is evidence that one of the mechanisms that ensure adequate blood flow to the brain involves the regulation of systemic arterial blood pressure (ABP) by an intracranial baroreceptor mechanism sensitive to changes in brain perfusion. This experimental animal study aimed to provide a detailed characterization of this mechanism. In studies conducted in anaesthetized and conscious rats, cerebral perfusion was experimentally manipulated by applying precise, incremental physiological changes in intracranial pressure (ICP). The data show that (i) the intracranial baroreceptor triggers robust, non-habituating, proportional sympathetic and cardiovascular responses to acute and repeated ICP increases within the physiological range; (ii) there is a linear relationship between systemic ABP and ICP, as well as between sympathetic nerve activity and ICP; (iii) decreases in brain partial pressure of oxygen induced by physiological changes in ICP are negligible, making the brain tissue hypoxia an unlikely cause of the evoked sympathetic and cardiovascular responses; (iv) ABP responses induced by decreased cerebral perfusion are restrained by inputs from arterial baroreceptors, but are unaffected by renal afferent activity; (v) intracranial baroreceptor mechanism contributes to sympathoexcitatory responses induced by acute arterial hypotension; and (vi) activation of the intracranial baroreceptor mechanism resets the arterial baroreflex centrally, allowing regulation of systemic blood pressure at a higher level required to counteract reduced brain perfusion. These data support the hypothesis that cerebral perfusion is a major determinant of sympathetic activity and systemic arterial blood pressure, regulated by the intracranial baroreceptor mechanism.

(Received 1 November 2024; accepted after revision 20 January 2025; first published online 10 February 2025)

**Corresponding authors** A. V. Gourine: Centre for Cardiovascular and Metabolic Neuroscience, Neuroscience, Physiology & Pharmacology, University College London, London WC1E 6BT, UK.    Email: a.gourine@ucl.ac.uk
N. Marina: Division of Medicine, University College London, London WC1E 6BT, UK.    Email: n.marina@ucl.ac.uk

**Key points**

- An intracranial baroreceptor mechanism contributes to the regulation of systemic arterial blood pressure to maintain cerebral blood flow.
- The intracranial baroreceptor mechanism triggers robust, non-habituating, proportional sympathetic and cardiovascular responses to physiological changes in brain perfusion.
- Arterial blood pressure increases induced by activation of the intracranial baroreceptor mechanism in response to reduced cerebral perfusion are restrained by inputs from arterial baroreceptors.
- The intracranial baroreceptor mechanism resets the arterial baroreflex centrally to regulate systemic blood pressure at a higher level required to counteract reduced brain perfusion.

# Introduction

Systemic arterial blood pressure (ABP), heart rate (HR) and cardiac contractility are controlled by neural circuits in the brainstem, which continually fine-tune sympathetic vasomotor and cardiac sympathetic and parasympathetic nerve activities in accord with the currently prevailing physiological and behavioural needs. This intricate

**Philippa Wittenberg** is a PhD student at University College London, supervised by Professor Alexander Gourine. She earned a master's degree from University College London, where she investigated the metabolic mechanisms underlying the regulation of cerebral blood flow. Her current research focuses on the central nervous mechanisms that regulate arterial blood pressure and sympathetic activity to maintain brain blood flow under conditions where cerebral perfusion might be compromised.

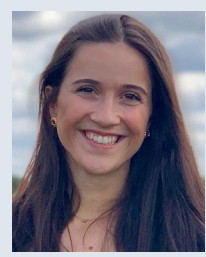

regulation depends on afferent information received by the brainstem autonomic control circuits from various sources. Baroreceptors, located in the carotid bifurcation and aortic arch, play an important role in the control of blood pressure and HR. In response to increases in ABP, these stretch-sensitive baroreceptor neurons (Zeng et al., 2018), with peripheral axons innervating the walls of the aortic arch and carotid sinus, initiate the arterial baroreflex, leading to reductions in HR, contractility and peripheral vascular resistance.

In addition to inputs from arterial baroreceptors and other afferents, the activities of the cardiovascular control circuits of the brainstem are modulated by changes in cerebral perfusion. Experimental studies conducted by Cushing (1901), and later by Rodbard and Stone (1955), first suggested the existence of an intracranial baroreceptor mechanism sensitive to decreases in blood flow to the brain. In these early investigations, performed in anaesthetized dogs, decreases in brain perfusion were induced experimentally by large (up to 200 mmHg) increases in intracranial pressure (ICP) and were found to trigger robust increases in systemic blood pressure. Results of more recent studies conducted in mice (Schmidt et al., 2018), rats (Marina et al., 2020), sheep (Guild et al., 2018; Vari et al., 2021) and humans (Schmidt et al., 2005, 2018) and involving smaller experimental manipulations of ICP, have provided further evidence supporting the existence of an intrinsic brain mechanism capable of sensing physiological changes in brain perfusion. Collectively, the data available in the literature suggest that when ICP increases and cerebral perfusion pressure (CPP) decreases, the intracranial baroreceptor mechanism triggers compensatory increases in sympathetic nerve activity and systemic ABP to restore and maintain cerebral blood flow, forming a homeostatic feedback loop. In this paper, we refer to this physiological mechanism as the 'intracranial baroreflex' (Schmidt et al., 2005).

Recent studies have identified brain glial cells, specifically astrocytes, as likely candidates for the role of intracranial baroreceptors (Marina et al., 2020). Evidence suggests that astrocytes are mechanosensitive (Bowser & Khakh, 2007; Turovsky et al., 2020; Yu et al., 2022) and respond to decreases in brain perfusion with increased $Ca^{2+}$ signalling (Marina et al., 2020). Blocking astrocyte signalling in the brainstem has been shown to reduce cardiovascular and sympathetic responses induced by changes in ICP (Marina et al., 2020; Turovsky et al., 2020). While these studies highlight the potential cellular mechanisms and dependence of sympathetic activity on brain perfusion, the functional operation of the intracranial baroreflex has not been systematically characterized.

In this experimental animal (rat) study, we applied precise physiological incremental cyclic increases in ICP

to: (i) describe the relationship between ICP, systemic ABP and sympathetic nerve activity; (ii) analyse the associated changes in the partial pressure of oxygen ($P_{O_2}$) and temperature within the sympathetic control areas of the rostral ventrolateral medulla (RVLM); (iii) characterize changes in sympathetic nerve activity synchronized with cardiac and respiratory cycles; (iv) investigate whether the blood pressure responses induced by activation of the intracranial baroreflex are modulated by inputs from the arterial baroreceptors and renal afferents; and (v) determine whether activation of the intracranial baroreceptor mechanism modulates the arterial baroreflex.

## Methods

### Animals and ethical approval

Acute experiments (Studies 1–4) were performed in male and female adult Sprague–Dawley rats (250–300 g) in accordance with the European Commission Directive 86/609/EEC (European Convention for the Protection of Vertebrate Animals used for Experimental and Other Scientific Purposes), and the UK Home Office Animals (Scientific Procedures) Act (1986) with project approval from the University College London Institutional Animal Care and Use Committee. Study 5, involving recordings in conscious animals, was performed in male Wistar rats (300–350 g) in accordance with the protocols approved by the University of Auckland Animal Ethics Committee (R25105). The animals were housed in temperature-controlled facilities (22–25°C) with a 12 h light–dark cycle (12 h:12 h, lights on at 07.00 h). Water and laboratory rodent food chow were provided *ad libitum*. This research was conducted in accordance with the animal ethics checklist outlined in *The Journal*'s instructions for authors.

### Experimental model (anaesthetized rats)

For Studies 1–4, rats were anaesthetized with urethane (induction: 1.3 g kg$^{-1}$, intraperitoneally; maintenance: 10–25 mg kg$^{-1}$ h$^{-1}$, intravenously) and instrumented for physiological recordings as described previously (Marina et al., 2020; Korsak et al., 2023). The femoral artery and vein were cannulated for measurements of ABP and administration of anaesthetic, respectively. The depth of anaesthesia was monitored and ensured by maintaining stable levels of ABP and HR and the absence of a response to a paw pinch. The trachea was cannulated, and the animal was ventilated mechanically with room air using a positive pressure ventilator (Model 683; Harvard Apparatus, Holliston, MA, USA) with a tidal volume of ~1 ml per 100 g of body weight and ventilator frequency

similar to the resting respiratory rate ($\sim$60 strokes min$^{-1}$). Tracheal pressure was recorded. The body temperature was monitored and maintained at 37.0 $\pm$ 0.5°C using a servo-controlled heating pad.

The animal was placed in a prone position and the head was secured in a stereotaxic frame. A dorsal midline incision was made to expose the surface of the skull, and two small craniotomies (diameter 0.6 mm;

coordinates: 1.0 mm caudal, 1.5 lateral, 4 mm ventral, from Bregma) were performed for cannulation of the lateral cerebral ventricles (LV). One ventricle was cannulated and connected via a saline-filled mini-catheter to a pressure transducer for recording ICP (Fig. 1*A*). Correct positioning of the cannula was confirmed by observing cardiac pulse- and respiratory cycle-related small oscillations in ICP. The contralateral ventricle was

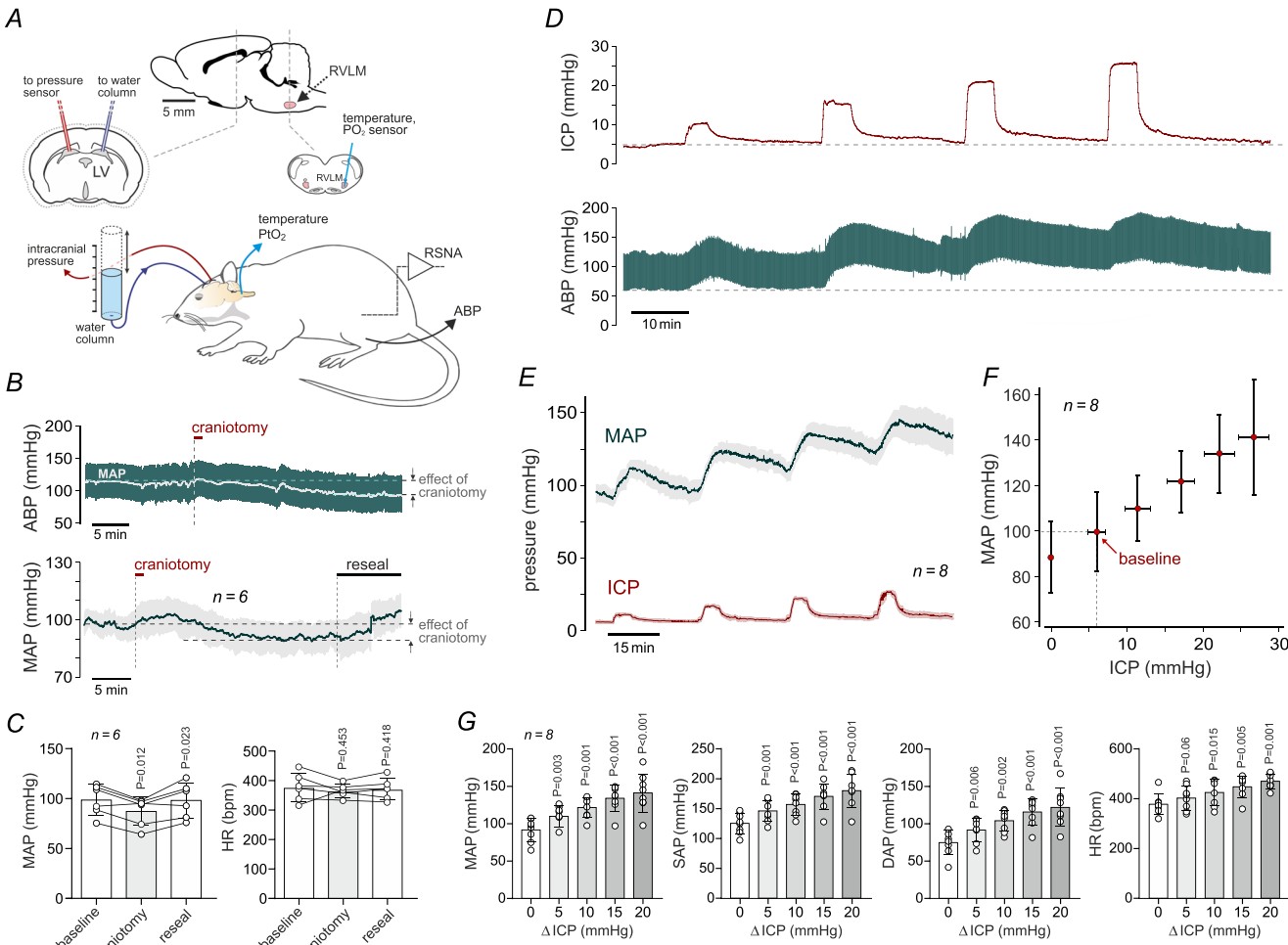

**Figure 1. Characterization of cardiovascular responses induced by activation of the intracranial baroreceptor mechanism in anaesthetized rats**

*A*, schematic illustration of the experimental model used in this study. In urethane-anaesthetized rats, the femoral artery was cannulated for recordings of arterial blood pressure (ABP); renal sympathetic nerve activity (RSNA) was recorded as a measure of central sympathetic drive. The lateral cerebral ventricles (LV) were cannulated and connected to a pressure transducer to record intracranial pressure (ICP) and to a 'water column' for experimental manipulation of ICP. Schematic drawings of the rat brain in parasagittal and coronal projections illustrate the cerebroventricular cannulae and the optical probe positioned in the brainstem to record partial pressure of oxygen ($P_{O_2}$) and temperature in the rostral ventrolateral medulla (RVLM). *B*, representative individual trace and averaged (means $\pm$ SEM) recordings showing changes in ABP and mean arterial blood pressure (MAP) following craniotomy (ICP = 0 mmHg) in anaesthetized rats. *C*, summary data illustrating the effects of craniotomy on MAP and heart rate (HR) (means $\pm$ SD). *D*, representative raw traces illustrating ABP responses to incremental cyclic increases in ICP in an anaesthetized rat. *E*, averaged recordings showing changes in MAP in response to incremental cyclic increases in ICP in anaesthetized rats. *F*, linear relationship between MAP and ICP ($r^2$ = 0.58, $P < 0.0001$) (means $\pm$ SD). *G*, summary data illustrating peak increases in MAP, systolic arterial blood pressure (SAP), diastolic arterial blood pressure (DAP) and HR in response to incremental cyclic increases in ICP by 5, 10, 15 and 20 mmHg (means $\pm$ SD) in anaesthetized rats. [Colour figure can be viewed at wileyonlinelibrary.com]

cannulated and connected via a saline-filled mini-catheter to a 50 ml reservoir (referred to here as a 'water column') for experimental manipulation of ICP (Fig. 1*A*). The cannulae were secured in place with cyanoacrylate adhesive.

### Electrophysiological recordings of phrenic nerve activity and renal sympathetic nerve activity

Phrenic nerve activity (PNA) and renal sympathetic nerve activity (RSNA) were recorded as measures of central respiratory and sympathetic drives, respectively. The nerve recordings were performed as described in detail previously (Gourine et al., 2008; Marina et al., 2011). The right phrenic nerve and left renal nerve were dissected and placed on bipolar silver wire electrodes. The signals were amplified ($\times 20,000$), filtered (80–1000 Hz) and sampled at a rate of 11 kHz. PNA and RSNA signals were rectified and smoothed (time constant = 100 ms). Changes in RSNA recorded during the experiment were normalized with respect to resting activity (100%) and complete absence of discharge (0%) following the administration of the ganglionic blocker hexamethonium (20 mg; i.v.), which was applied at the end of each experiment. Peak systolic blood pressure-triggered (BP-T RSNA) and peak tracheal pressure (TP)-triggered (TP-T RSNA) average wave-forms of rectified and smoothed RSNA were generated using Spike2 software (Cambridge Electronic Design, Cambridge, UK). As blood pressure was recorded from the femoral artery, the BP-T RSNA average waveforms were offset by 32 ms to account for the pressure wave propagation delay (measured in a separate experiment involving simultaneous recordings of ABP in the femoral artery and the aorta).

### Recordings of partial pressure of oxygen and temperature in the brainstem

A small craniotomy was performed in the occipital bone at 11.5 mm caudal from Bregma and 2 mm lateral from the midline. $P_{O_2}$ and temperature in the brainstem region of the RVLM were recorded using a combined optical oxygen and temperature bare-fibre sensor (250 μm tip diameter; OxyLite system; Oxford Optronix, Adderbury, UK) placed 8.5–9.0 mm below the surface of the skull (Fig. 1*A*). The operation of the oxygen sensor is based on an optical fluorescence method that allows real-time recording of absolute changes in tissue $P_{O_2}$.

### Recordings of arterial blood pressure in conscious rats

Rats ($n = 11$) underwent surgery to implant radio-telemetry devices for simultaneous recordings of ABP and ICP (TRM54PP; ADInstruments, Oxford, UK),

as well as brain tissue oxygen ($P_{O_2}$) using amperometry (TR57Y). The oxygen sensor was calibrated prior to implantation, by measuring the voltage output in a saline solution saturated with 100% $N_2$, 10%, 21% or 100% $O_2$, equating to $O_2$ concentrations of 0, 114, 240 and 1260 μM, respectively, as described previously (Russell et al., 2012). The telemetry devices were sterilized with a 2% glutaraldehyde solution for 2–4 h and then rinsed thoroughly with sterile saline before implantation. The animals were anaesthetized with 2–4% isoflurane in 100% oxygen. Antibiotic (enrofloxacin) and analgesia (buprenorphine) were administered subcutaneously. Using an aseptic technique, pressure catheters were implanted into the abdominal aorta and subdural space to record ABP and ICP, respectively, as described (Guild et al., 2015; Fong et al., 2021). During the same surgery, a carbon paste electrode (CPE) reference and auxiliary electrode were inserted into the cortex, with the CPE placed 4.5 mm lateral from Bregma and 2.3 mm below the surface of the brain. An intracerebroventricular cannula was implanted into the left lateral ventricle, and externalized via an intrascapular port. Cranial electrodes and cannula were secured in place using stainless steel surgical screws and dental cement, with care taken to ensure that the integrity of the intracranial space was maintained. Telemetry transducers were secured in the abdomen. Animals were given at least 7 days to recover after the surgery before the main experiments.

### Experimental protocols

**Study 1: cardiovascular and sympathetic responses to incremental increases in intracranial pressure.** The rats were anaesthetized, mechanically ventilated and instrumented for physiological recordings, as described above. In these experiments, the effects of acute reductions in ICP were first assessed by recording ABP and HR for 15 min before and 30 min after craniotomy (ICP = 0 mmHg) and then for 15 min after the restoration of basal ICP (~6 mmHg) following the implantation of the intraventricular cannulae. Incremental cyclic increases in ICP were induced by changing the vertical position of the water column connected via the implanted cannula and saline-filled catheter to the cerebral ventricular system. Changes in ABP, HR, brainstem tissue $P_{O_2}$, brainstem temperature, RSNA and PNA evoked by increases in ICP by 5, 10, 15 and/or 20 mmHg (representing changes in ICP within the physiological range; Petersen et al., 2016) were recorded. Each level of increased ICP was maintained for 5 min, followed by 20 min recovery period to allow ICP to return to the baseline (Fig. 1*D*).

**Study 2: cardiovascular responses to changes in intracranial pressure under conditions of arterial baroreceptor**

**or renal denervation.** The rats were anaesthetized, mechanically ventilated and instrumented for recordings of systemic ABP, ICP and experimental manipulation of ICP, as described above. In one group of animals, unilateral arterial baroreceptor and cardiopulmonary deafferentation was performed by transection of the left vagus nerve, left carotid sinus nerve and left aortic depressor nerve (ADN) (unilateral arterial baroreceptor denervation), followed by the assessment of the cardiovascular responses induced by increased ICP to 10 mmHg. After a recovery period of up to 1 h to allow ABP and HR to return to the baseline, complete peripheral baroreceptor denervation was achieved by applying lidocaine (1%) to the vagus nerve, carotid sinus nerve and ADN on the right side, followed by the assessment of the cardiovascular response induced by increased ICP (10 mmHg).

In a separate experimental group of animals, renal nerve bundles supplying the left kidney were dissected and cut (unilateral renal denervation) followed by the assessment of the cardiovascular response induced by increased ICP to 10 mmHg. After a recovery period of up to 1 h, complete renal denervation was achieved by the application of lidocaine (1%) to the renal nerves supplying the right kidney, followed by the assessment of the cardiovascular response to increased ICP (10 mmHg).

**Study 3: sympathetic responses to acute arterial hypotension under conditions of peripheral baroreceptor denervation.** The rats were anaesthetized, mechanically ventilated and instrumented for recordings of systemic ABP and RSNA, as described above. Acute arterial hypotension was induced by intravenous infusion of sodium nitroprusside (10 μg kg$^{-1}$ min$^{-1}$; 2 min). Complete arterial baroreceptor and cardiopulmonary deafferentation was produced as described above, but innervation of the kidneys was left intact in this experiment. The HR and sympathetic responses to acute arterial hypotension induced by sodium nitroprusside were assessed in the presence and absence of the peripheral arterial baroreceptor input.

**Study 4: arterial baroreflex sensitivity under conditions of increased intracranial pressure.** The rats were anaesthetized, mechanically ventilated and instrumented for recordings of systemic ABP, ICP and experimental manipulation of ICP. Arterial baroreflex sensitivity was assessed by analysing changes in ABP and HR in response to electrical stimulation of the ADN. The right ADN was dissected, cleared of connective tissue, placed on a bipolar silver wire electrode, and secured in place with silicone impression material. Baseline ABP and HR were recorded for 10 min at resting ICP conditions. This was followed by three consecutive electrical stimulations of the ADN (pulse duration 0.1 ms; intensity 0.5 mA) for 10 s at frequencies of 1, 5 and 10 Hz, repeated at 3-min intervals between stimulations. The series of ADN stimulations were repeated under conditions of increased ICP, first by 10 mmHg and then by 15 mmHg above the baseline level.

**Study 5: cardiovascular responses to increases in intracranial pressure in conscious rats.** During the experiment, the ICP cannula was connected via the intrascapular port to an infusion pump. Once the rat was quiescent, sterile artificial cerebrospinal fluid (aCSF) was infused into the lateral cerebral ventricle for 45 min at a stepwise infusion rate of 5, 10 and 15 μl min$^{-1}$ (15 min per step). On completion of the infusion, the cannula was opened to the atmosphere, then the port was capped and the animal returned to its home cage.

At the end of the experiments, the animals were humanely killed with an overdose of anaesthetic (pentobarbital sodium 200 mg kg$^{-1}$, i.v.; Animalcare, York, UK).

## Data analysis

Physiological data were acquired using a Power1401 interface and analysed offline using Spike2 software (Cambridge Electronic Design). Changes in ABP, HR, TP, brain tissue $P_{O_2}$ and temperature, RSNA and PNA induced by changes in ICP were compared using Student's $t$ test, two-way ANOVA, or the Wilcoxon signed-rank test, as appropriate. Data are reported in the text as means ± SD and are illustrated in the figures as individual values and/or means ± SD. For presentation purposes, traces of averaged recordings of mean arterial blood pressure (MAP), ICP, HR and RSNA (Figs 1*B*, 1*E*, 4*B*, 4*D* and 5*B*) are shown as means ± SEM. Differences with $P < 0.05$ were considered significant.

## Results

### Study 1: cardiovascular and sympathetic responses to incremental increases in intracranial pressure

In urethane-anaesthetized rats, resting (basal) ICP was recorded at 6.2 ± 2.0 mmHg ($n = 8$), which was comparable to ICP values measured in conscious, freely behaving rats (Guild et al., 2015) (also see results of Study 5). Acute reduction of ICP to 0 mmHg following craniotomy resulted in a decrease in mean ABP by 11 ± 7 mmHg ($P = 0.012$, $n = 6$) (Fig. 1*B* and *C*). Restoration of basal ICP following the implantation of intraventricular cannulae and repair increased ABP back to the baseline level recorded before craniotomy (increase by 11 ± 8 mmHg; $P = 0.023$) (Fig. 1*B* and *C*). These experimental manipulations (craniotomy and repair) had no effect on HR ($P = 0.453$; Fig. 1*C*).

It was next found that in this experimental model, incremental cyclic increases in ICP led to significant changes in ABP and HR, which increased in proportion to the strength of the stimulus (Fig. 1*D–G*). In a study group of eight male rats, we observed that experimentally induced increases in ICP by 5, 10, 15 and 20 mmHg increased mean ABP by $18 \pm 12$, $30 \pm 16$, $42 \pm 18$ and $50 \pm 23$ mmHg, and HR by $25 \pm 31$, $47 \pm 42$, $70 \pm 49$ and $92 \pm 48$ bpm, respectively. The blood pressure responses followed increases in ICP with a mean delay of $22 \pm 5$ s ($n = 8$), consistent with the previously reported data (Marina et al., 2020). A linear relationship between mean ABP and ICP within the range of 0–30 mmHg was observed ($r^2 = 0.58$, $P < 0.001$; Fig. 1*F*). Peak increases in MAP in response to cyclic increases in ICP were not significantly different between male and female animals. In male rats ($n = 8$), experimentally induced increases in ICP by 5, 10, 15 and 20 mmHg raised MAP to $110 \pm 14$, $122 \pm 14$, $134 \pm 17$ and $141 \pm 25$ mmHg, respectively. In a representative group of six female rats, the corresponding increases in ICP raised MAP to $125 \pm 24$, $129 \pm 30$, $128 \pm 27$ and $132 \pm 28$ mmHg, respectively.

Recordings of $P_{tO_2}$ and temperature in the RVLM showed that acute increases in ICP induced small transient decreases in both $P_{tO_2}$ and temperature, referred to in Fig. 2*A–C* as 'initial dips'. In response to ICP increases of 15 and 20 mmHg, RVLM $P_{tO_2}$ decreased by $1.4 \pm 1.1$ mmHg ($P = 0.031$) and $1.7 \pm 0.9$ mmHg ($P = 0.007$), while temperature decreased by $0.06 \pm 0.05$°C ($P = 0.020$) and $0.09 \pm 0.08$°C ($P = 0.037$), respectively. These initial dips were followed by significant increases (overshoots) in both brainstem tissue $P_{O_2}$ (by $10.0 \pm 7.6$ mmHg, $P = 0.024$, and $12.1 \pm 9.5$ mmHg, $P = 0.027$) and temperature (by $0.48 \pm 0.34$°C, $P = 0.017$, and $0.61 \pm 0.26$°C, $P = 0.002$, respectively) (Fig. 2*A–C*).

Sympathetic responses to changes in ICP were analysed next. It was found that increases in ICP by 5, 10, 15 and 20 mmHg triggered robust increases in mean RSNA by $21 \pm 23\%$ ($P = 0.055$), $58 \pm 45\%$ ($P = 0.015$), $103 \pm 67\%$ ($P = 0.007$) and $172 \pm 92\%$ ($P = 0.003$), respectively ($n = 7$) (Fig. 3*A–E*), supporting the hypothesis that vasomotor sympathetic activity is strongly dependent on ICP (and, therefore, cerebral perfusion), as previously suggested (Schmidt et al., 2018). In a series of experiments involving simultaneous recordings of RSNA and PNA, increases in ICP were found to enhance central respiratory drive (Fig. 3*A*, *B*, *D* and *F*). Analysis of blood pressure-triggered RSNA waveforms showed strong inhibition of sympathetic discharge during systole, under both basal and raised ICP conditions, and demonstrated that increases in ICP amplified bursts of sympathetic activity during the diastolic phase of the cardiac cycle (Fig. 3*C* and *F*). These bursts were largest during the inspiratory phase of the respiratory cycle (Fig. 3*B*, *D* and *F*). Coupling between the respiratory activity (PNA)

and RSNA was not affected under conditions of increased ICP.

## Study 2: cardiovascular responses to changes in intracranial pressure under conditions of arterial baroreceptor or renal denervation

In rats subjected to unilateral arterial baroreceptor and cardiopulmonary denervation – achieved by transecting the left vagus nerve, left carotid sinus nerve and left ADN – the ABP and HR responses induced by 10 mmHg increases in ICP were similar to the responses recorded in intact animals (Fig. 4*C*). Complete arterial baroreceptor denervation combined with bilateral vagotomy, following the blockade of action potential propagation in the vagus, carotid sinus and aortic nerves on the right side (with lidocaine), increased ABP and HR, as expected (Fig. 4*B* and *D*). Arterial baroreceptor denervation was confirmed by the absence of an HR response to a pressor stimulus (noradrenaline; 0.1 μg kg$^{-1}$, I.V.; Fig. 4*A*).

Under conditions of bilateral arterial baroreceptor and cardiopulmonary denervation, mean ABP increased by $44 \pm 18$ mmHg ($n = 10$) in response to a 10 mmHg increase in ICP. This response was greater ($P = 0.027$) than the increase in mean ABP induced by the same ICP stimulus applied under conditions of unilateral denervation (by $28 \pm 14$ mmHg) (Fig. 4*B* and *C*). HR responses were unaffected by baroreceptor denervation (Fig. 4*C*). Cardiovascular responses to increases in ICP were unaffected under conditions of unilateral or bilateral denervation of the kidneys ($n = 6$) (Fig. 4*B* and *C*).

## Study 3: sympathetic responses to acute arterial hypotension under conditions of peripheral baroreceptor denervation

Acute arterial hypotension was induced by intravenous infusion of sodium nitroprusside (10 μg kg$^{-1}$ min$^{-1}$; 2 min) (Fig. 4*D*). As expected, in animals with intact arterial baroreceptors, the sodium nitroprusside-induced decrease in blood pressure (mean change from $86 \pm 12$ mmHg to $45 \pm 6$ mmHg) triggered compensatory increases in RSNA (by $46 \pm 30\%$; $n = 6$; $P = 0.013$) and HR (by $29 \pm 16$ bpm; $n = 11$; $P < 0.001$) (Fig. 4*D*). In this group of animals (without craniotomy), bilateral arterial baroreceptor and cardiopulmonary denervation led to an elevation of baseline ABP (by $12 \pm 15$ mmHg; $P = 0.021$) and HR (by $65 \pm 49$ bpm; $P = 0.001$) (Fig. 4*D*). However, the sympathetic response (increase in RSNA by $38 \pm 33\%$; $P = 0.029$) and HR response (increase by $23 \pm 13$ bpm; $P < 0.001$) to acute arterial hypotension induced by sodium nitroprusside (mean decrease to $44 \pm 7$ mmHg) were largely preserved (Fig. 4*D*). There were no differences in the peak increases in RSNA and HR ($P = 0.134$ and $P$

= 0.334, respectively) before and after bilateral arterial baroreceptor and cardiopulmonary denervation.

## Study 4: arterial baroreflex sensitivity under conditions of increased intracranial pressure

Arterial baroreflex sensitivity was assessed by analysing changes in ABP and HR in response to electrical stimulation of ADN at resting ICP and under conditions of increased ICP (by 10 and 15 mmHg) in a separate study group of 10 rats. Under resting ICP conditions, ADN stimulation applied at a frequency of 1 Hz decreased mean

ABP by $5 \pm 6$ mmHg ($P = 0.009$) and HR by $12 \pm 8$ bpm ($P < 0.001$) (Fig. 5*B* and *C*). ADN stimulation applied at 1 Hz under conditions of increased ICP had no significant effect on ABP and HR (Fig. 5*B* and *C*). ADN stimulation at 5 Hz decreased mean ABP by $23 \pm 10$ mmHg ($P < 0.001$) and HR by $43 \pm 31$ bpm ($P = 0.003$). The blood pressure decreases in response to ADN stimulation at 5 Hz were significantly smaller under conditions of raised ICP (Fig. 5*B* and *C*). There were no significant differences in HR responses to 5 Hz ADN stimulation under baseline and increased ICP conditions (Fig. 5*B* and *C*). ADN stimulation at 10 Hz decreased mean ABP

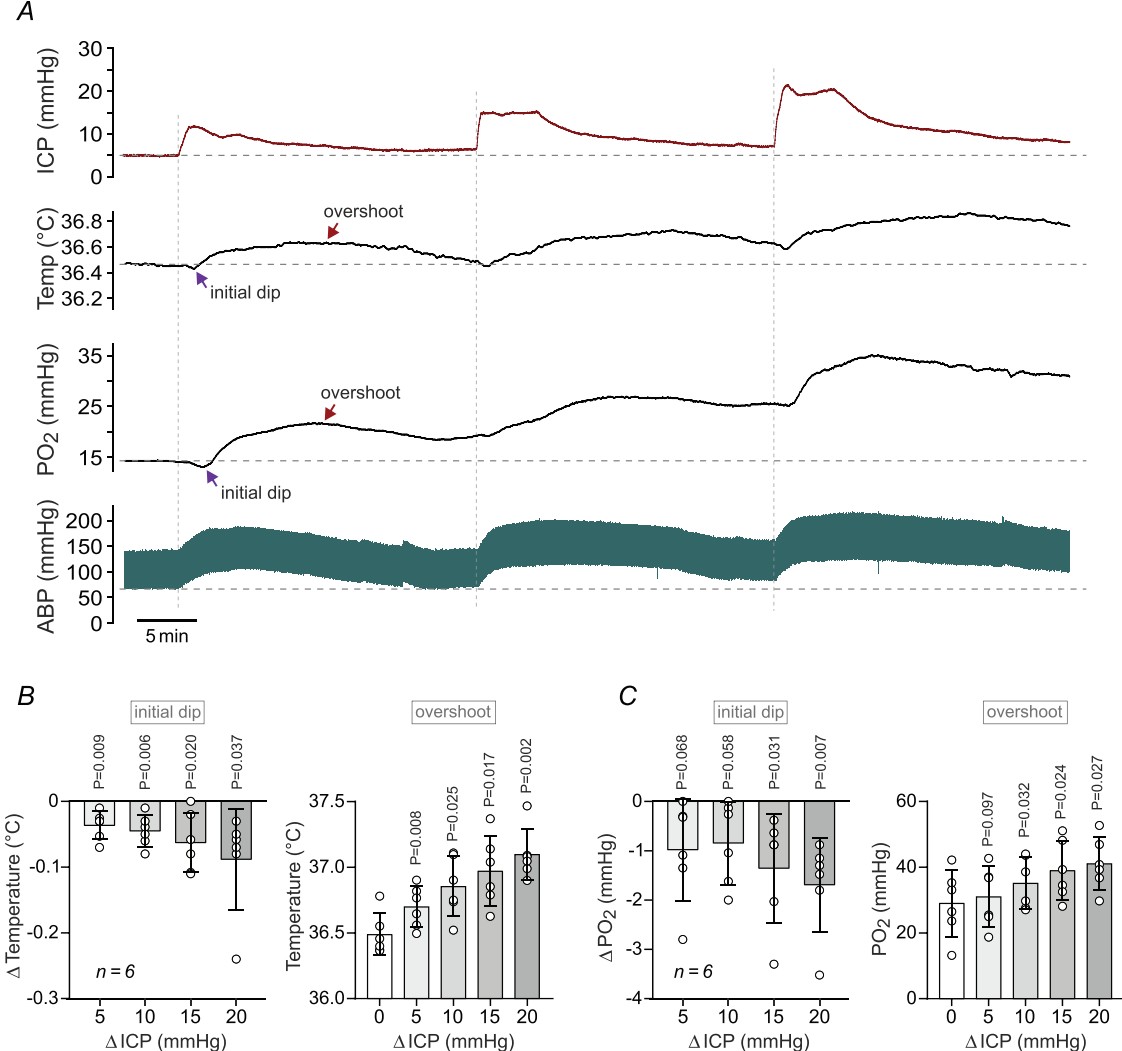

**Figure 2. Effects of intracranial pressure increases on brainstem $P_{O_2}$ and temperature in anaesthetized rats**
*A*, representative raw traces illustrating changes in arterial blood pressure (ABP), partial pressure of oxygen ($P_{O_2}$) and temperature in the rostral ventrolateral medulla in response to incremental cyclic increases in intracranial pressure (ICP) in an anaesthetized rat. *B*, summary data showing the magnitude of the initial decreases (initial dips) in RVLM temperature and the subsequent increases (overshoots) in temperature in response to incremental cyclic increases in ICP by 5, 10, 15 and 20 mmHg in anaesthetized rats (means $\pm$ SD). *C*, summary data illustrating the magnitude of the initial decreases in RVLM $P_{O_2}$ and the subsequent overshoots in $P_{O_2}$ in response to incremental cyclic increases in ICP (means $\pm$ SD). [Colour figure can be viewed at wileyonlinelibrary.com]

by $28 \pm 8$ mmHg ($P < 0.001$) and HR by $52 \pm 36$ bpm ($P < 0.001$). There were no significant differences in ABP and HR responses to 10 Hz ADN stimulation under baseline and increased ICP conditions (Fig. 5*B* and *C*). The input (frequency of ADN stimulation)–output ($\Delta$MAP, $\Delta$HR) relationships for the baroreceptor reflex (Dampney, 2017) are illustrated in Fig. 5*C*.

### Study 5: cardiovascular responses to incremental increases in intracranial pressure in conscious rats

In conscious rats ($n = 11$), the mean resting levels of ICP and MAP were $5.4 \pm 2.2$ mmHg and $100 \pm 8$ mmHg, respectively. During intracerebroventricular aCSF infusion, ICP increased to $13.3 \pm 2.7$ mmHg, accompanied by an increase in MAP to $110 \pm 12$ mmHg. The blood pressure response induced by activation of the intracranial baroreceptor mechanism effectively maintained the stability of cerebral perfusion and cortical tissue $P_{O_2}$ (Fig. 6). The study showed a strong linear relationship between MAP and ICP in conscious rats ($R^2 = 0.83$, $P < 0.001$; Fig. 6*B*).

### Discussion

There is significant experimental evidence supporting the hypothesis that one of the mechanisms regulating cerebral blood flow involves specialized intracranial baroreceptors

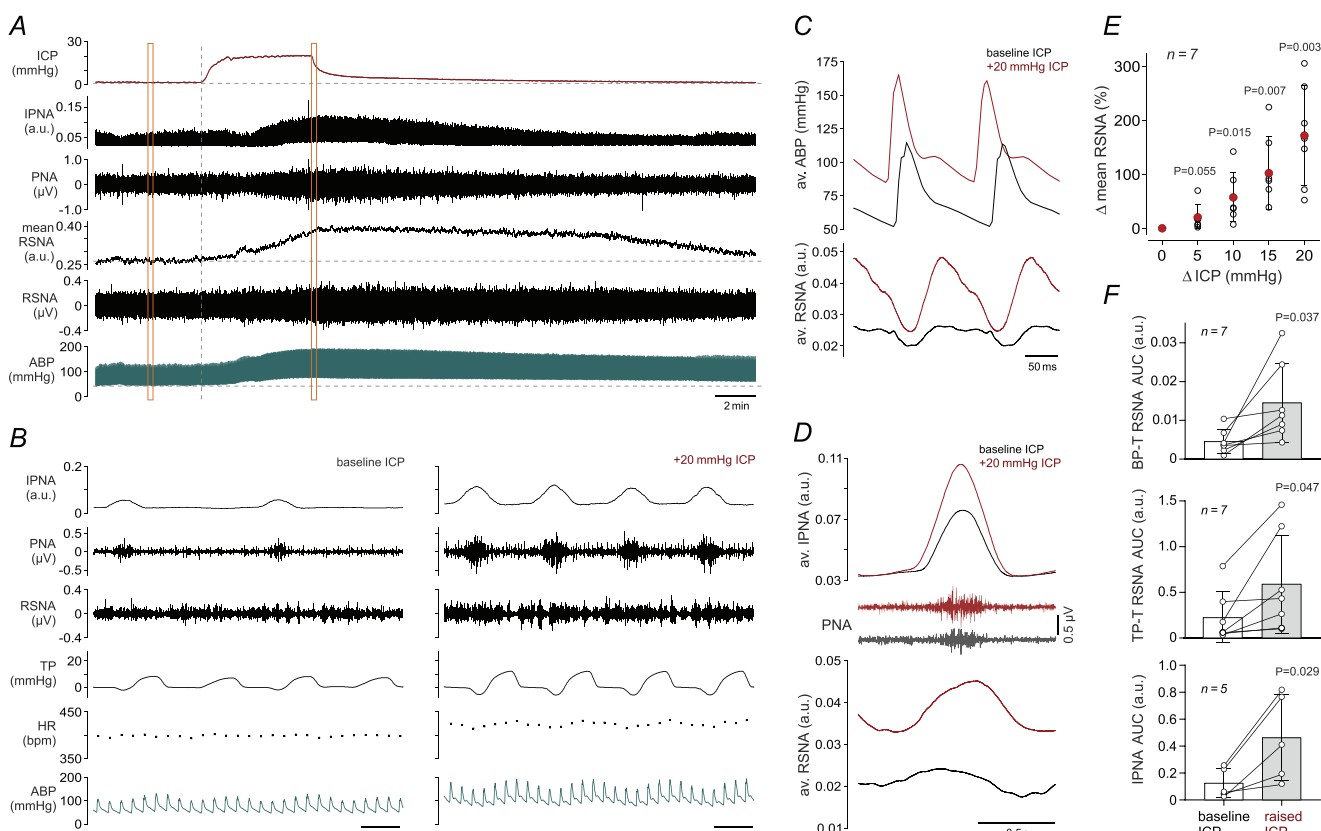

**Figure 3. Effects of intracranial pressure increases on sympathetic nerve activity and central respiratory drive in anaesthetized rats**

*A*, representative time-condensed traces illustrating recordings of phrenic nerve activity (PNA), renal sympathetic nerve activity (RSNA) and arterial blood pressure (ABP) before, during and after a 5 min period of experimentally raised intracranial pressure (ICP) in an anaesthetized rat. IPNA, integrated phrenic nerve activity. *B*, expanded traces showing recordings of IPNA, PNA, RSNA, tracheal pressure (TP), heart rate (HR) and ABP at baseline ICP and during the period of increased ICP. *C*, averaged (av.) ABP waveforms and ABP-triggered integrated RSNA waveforms recorded at baseline conditions and at the peak of the response to increased ICP, expressed in arbitrary units (a.u.). *D*, averaged (av.) integrated PNA waveforms, PNA-triggered RSNA waveforms and examples of raw PNA activity recorded at baseline conditions and at the peak of the response induced by increased ICP, expressed in arbitrary units (a.u.). *E*, summary data illustrating percentage increases in RSNA in response to incremental increases in ICP by 5, 10, 15 and 20 mmHg in anaesthetized rats (means $\pm$ SD). *F*, summary data showing the areas under the curve (AUC) of blood pressure (BP)- and tracheal pressure (TP)-triggered RSNA waveforms (BP-T RSNA AUC and TP-T RSNA AUC, respectively) and the IPNA AUC recorded at baseline conditions and at the peak of the response to increased ICP in anaesthetized rats (means $\pm$ SD). [Colour figure can be viewed at wileyonlinelibrary.com]

sensitive to changes in brain perfusion (Cushing, 1901; Guild et al., 2018; Hoff & Reis, 1970; Marina et al., 2020; Rodbard & Stone, 1955; Schmidt et al., 2005, 2018; Vari et al., 2021). When CPP drops, this mechanism triggers compensatory increases in sympathetic nerve activity and systemic ABP to restore and maintain brain blood flow. This experimental animal study was designed to characterize the operation of the intracranial baroreflex,

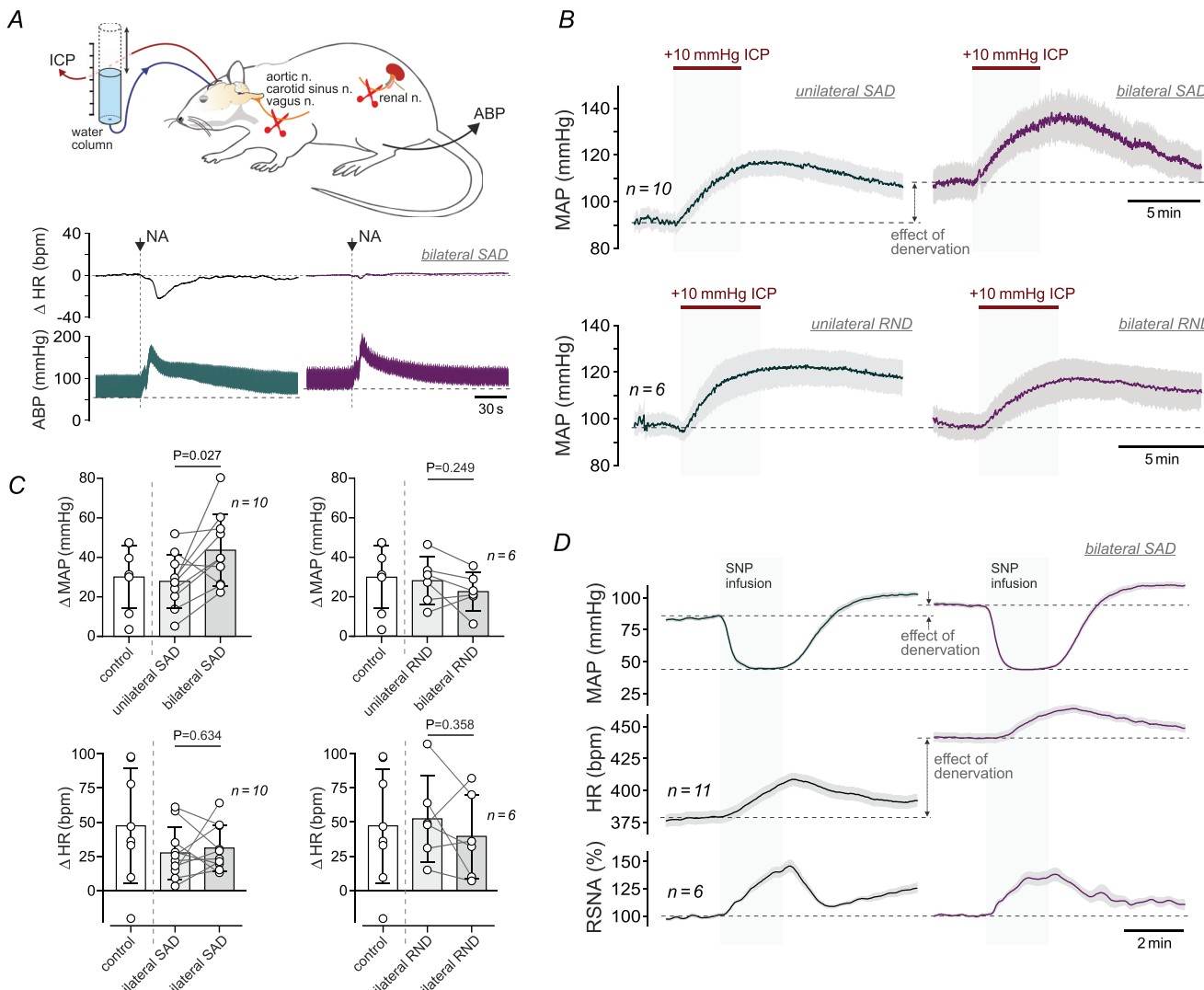

**Figure 4. Characterization of cardiovascular and sympathetic responses induced by activation of the intracranial baroreceptor mechanism under conditions of sino-aortic denervation (SAD) or renal denervation (RND) in anaesthetized rats**

*A*, schematic illustration of the experimental approach. In one group of animals, unilateral arterial baroreceptor and cardiopulmonary denervation was performed by transection of the left vagus nerve, left carotid sinus nerve and left aortic depressor nerve (unilateral SAD). Bilateral SAD was achieved by applying lidocaine (1%) to the vagus nerve, carotid sinus nerve and aortic depressor nerve on the contralateral side. In a separate group of animals, renal nerve bundles supplying the left kidney were dissected and cut (unilateral RND); bilateral RND was achieved by applying lidocaine to the renal nerves supplying the right kidney. Arterial baroreceptor denervation was confirmed by the absence of a heart rate (HR) response to a pressor stimulus (noradrenaline, NA; 0.1 µg kg$^{-1}$, i.v.). *B*, averaged recordings of mean arterial blood pressure (MAP) illustrating responses to increased intracranial pressure (ICP) under conditions of unilateral or bilateral SAD or RND in anaesthetized rats (means ± SEM). *C*, summary data of peak increases in MAP and HR in response to 10 mmHg increases in ICP under control conditions (data from Study 1) and under conditions of unilateral or bilateral SAD or RND (means ± SD). *D*, averaged traces illustrating HR and renal sympathetic nerve activity (RSNA) responses to acute arterial hypotension induced by infusion of sodium nitroprusside (SNP; 10 µg min$^{-1}$) before and after bilateral SAD and cardiopulmonary deafferentation in anaesthetized rats (means ± SEM). [Colour figure can be viewed at wileyonlinelibrary.com]

focusing on the specifics of sympathetic responses to decreases in cerebral perfusion and the regulation of blood pressure by the intracranial baroreceptor mechanism in relation to changes in peripheral arterial baroreceptor activity.

Our studies of the cardiovascular and sympathetic responses to incremental cyclic increases in ICP revealed a striking linear relationship between MAP and ICP (within the physiological range of ICP changes) in both anaesthetized and conscious rats (Figs 1*F* and 6*B*). We also report a corresponding linear relationship between sympathetic nerve activity and ICP (Fig. 3*E*), an observation supported by the results of previous studies conducted in sheep and humans (Guild et al., 2018; Schmidt et al., 2018). The intracranial baroreflex triggers robust, non-habituating responses to acute and repeated increases in ICP. An interesting observation was that when ICP decreased to 0 mmHg following acute craniotomy, it was followed by a small but significant reduction in ABP,

which stabilized at a level ~10 mmHg below baseline. This finding is consistent with the results of clinical studies that reported decreases in ABP following decompressive craniectomy in patients with traumatic brain injury or other pathologies (Bharath et al., 2020; Jo et al., 2021; Timofeev et al., 2008). Collectively, the data obtained in the current study using anaesthetized and conscious rat models are in full agreement with previously reported experimental and clinical evidence and strongly support the hypothesis that ICP, and consequently cerebral perfusion, are major determinants of sympathetic nerve activity and ABP.

It could be argued that the sympathetic and blood pressure responses to increased ICP are driven by brain tissue hypoxia and the hypoxic activation of pre-sympathetic RVLM neurons, which provide the major excitatory drive to spinal sympathetic pre-ganglionic neurons. Indeed, there is strong evidence that sympathoexcitatory RVLM neurons and brain glial

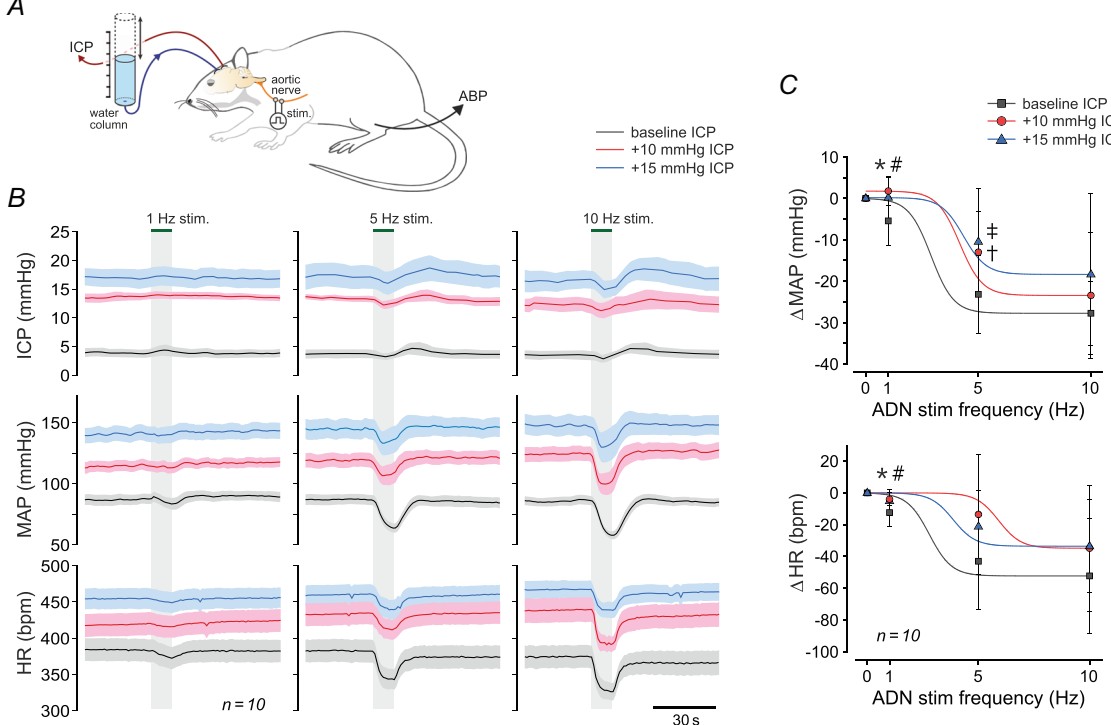

**Figure 5. Arterial baroreflex sensitivity at different levels of intracranial pressure**
*A*, schematic illustration of the experimental approach to study the cardiovascular responses to electrical stimulation of the aortic depressor nerve (ADN) at baseline intracranial pressure (ICP) and under conditions of increased ICP in anaesthetized rats. *B*, averaged recordings of ICP, mean arterial blood pressure (MAP) and heart rate (HR) illustrating responses induced by electrical stimulation of the ADN at frequencies of 1, 5 and 10 Hz under baseline ICP conditions and at the peak of the response to increased ICP (by 10 and 15 mmHg) in anaesthetized rats (means ± SEM). *C*, summary data showing peak changes in MAP and HR in response to ADN stimulation (means ± SD); sigmoidal fitting illustrates the input (frequency of ADN stimulation)–output (ΔMAP, ΔHR) relationship for the baroreceptor reflex at different levels of ICP. *$P < 0.001$ and #$P = 0.007$ for differences in MAP responses at baseline ICP compared to conditions of increased ICP by 10 and 15 mmHg, respectively; ‡$P = 0.030$ and †$P = 0.007$ for differences in MAP responses at baseline ICP compared to conditions of increased ICP by 10 and 15 mmHg, respectively; *$P = 0.005$ and #$P = 0.017$ for differences in HR responses at baseline ICP compared to conditions of increased ICP by 10 and 15 mmHg, respectively. [Colour figure can be viewed at wileyonlinelibrary.com]

cells are sensitive to hypoxia (Angelova et al., 2015; Christie et al., 2023; Marina et al., 2015; Sun & Reis, 1993, 1994). In this study, we explored whether increases in ICP (by up to 20 mmHg) could reduce cerebral blood flow sufficiently to impair brain tissue perfusion and cause brainstem hypoxia. Oxygen measurements in the RVLM showed that such increases in ICP led to small (by ~1.5 mmHg) and transient initial decreases in $P_{tO_2}$, followed by significant (by up to 12 at 20 mmHg ICP) overshoots that, with some delay, mirrored changes in systemic blood pressure. Simultaneous measurements

of temperature showed similar profiles of brainstem tissue temperature changes, with initial small decreases of ~0.1°C, followed by delayed increases (by ~0.5°C), with temporal profiles that aligned with changes in ABP and could potentially be explained by increased metabolic activity of the pools of pre-sympathetic neurons in the area. These data suggest that changes in ICP within the physiological range, as applied in the experiments of this study, are not associated with brain tissue hypoxia, and therefore, the resulting sympathetic and cardiovascular

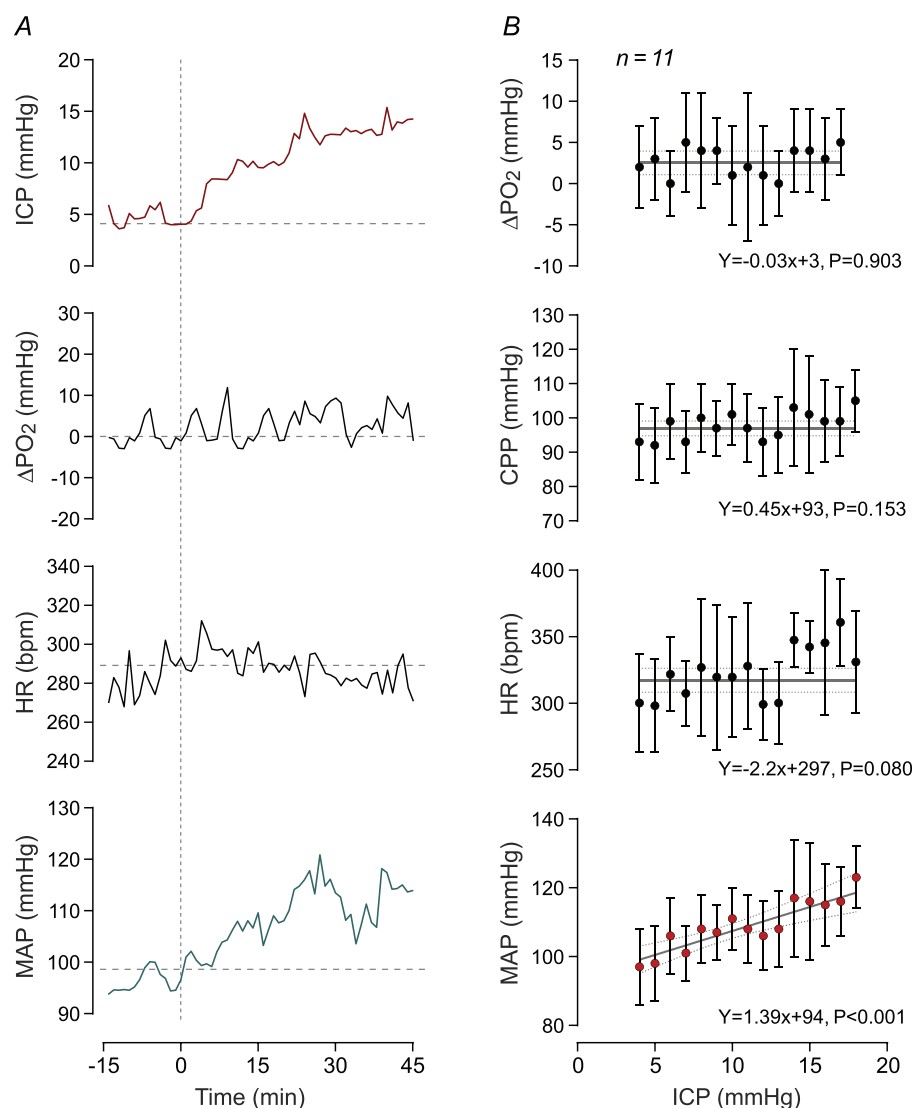

**Figure 6. The intracranial baroreceptor mechanism maintains cerebral perfusion under conditions of increased intracranial pressure in conscious rats**

*A*, representative traces illustrating changes in intracranial pressure (ICP), partial pressure of $O_2$ ($P_{O_2}$) in the cerebral cortex, heart rate (HR) and mean arterial blood pressure (MAP) in response to intracerebroventricular infusion of artificial cerebrospinal fluid in a conscious rat. *B*, summary data obtained in conscious rats illustrating averaged values of cortical tissue $P_{O_2}$, cerebral perfusion pressure (CPP), HR and MAP binned by 1 mmHg increments in ICP (means ± SD). Linear regression equations are provided, testing for non-zero slopes. Increases in MAP showed a significant positive linear relationship with increasing ICP, which maintained constant cerebral perfusion and brain tissue $P_{O_2}$. [Colour figure can be viewed at wileyonlinelibrary.com]

responses are driven by stimuli other than changes in oxygen delivery to the brainstem.

Simultaneous recordings of phrenic and renal nerve activities showed that activation of the intracranial baroreflex by raised ICP enhances central respiratory drive and increases the amplitude of rhythmic bursts of sympathetic activity, with peaks of discharge occurring during the diastolic phase of the cardiac cycle. More specifically, at elevated ICP, the amplitude of cardiac-cycle-related bursts of sympathetic activity was found to be largest during the inspiratory phase of the respiratory cycle, suggesting that activation of the intracranial baroreflex amplifies the excitatory drive from the respiratory network to pre-sympathetic neurons (Moraes et al., 2014). Renal nerve discharge remained rhythmic, with the lowest level of activity during the systolic phase of the cardiac cycle, indicating that vasomotor sympathetic activity is strongly modulated by the baroreflex at all levels of ICP. This observation prompted us to investigate how inputs from arterial baroreceptors modulate cardiovascular responses induced by activation of the intracranial baroreflex.

The central nervous pathway of the arterial baroreflex includes an inhibitory synaptic connection between neurons of the caudal ventrolateral medulla (which receive excitatory inputs from arterial baroreceptors via the nucleus of the solitary tract) and sympathoexcitatory neurons of the RVLM. This neural pathway is ultimately responsible for inhibiting vasomotor and cardiac sympathetic activities in response to acute increases in arterial baroreceptor afferent discharge (Dampney, 2017; Spyer, 1994). The robust increases in blood pressure in response to elevated ICP suggest that the potent inhibitory influence of increased arterial baroreceptor activity may be overridden or blocked by the intracranial baroreceptor mechanism. To test this hypothesis, we compared the cardiovascular responses to increases in ICP in the presence and absence of arterial baroreceptor input.

It was found that cardiovascular responses induced by increased ICP were unaffected by unilateral transection of the aortic nerve, carotid sinus nerve and vagus nerve. However, blood pressure responses to the same increases in ICP were significantly enhanced after subsequent anaesthetic blockade of the same set of nerves on the contralateral side (bilateral deafferentation). These results suggested that the arterial baroreflex continues to regulate blood pressure at elevated ICP and restrains the blood pressure increases in response to activation of the intracranial baroreceptor mechanism.

It is often assumed that sympathetic responses to acute decreases in ABP are mediated through the same peripheral baroreceptor reflex pathway, where arterial baroreceptor 'unloading' leads to disinhibition of sympathetic activity, increasing cardiac output and peripheral vascular resistance. Data obtained in animals with bilateral arterial baroreceptor denervation (and vagotomy), illustrated by Fig. 4*D*, suggest that the intracranial baroreceptor mechanism is responsible, at least in part, for the sympathoexcitatory responses to acute falls in blood pressure, independently of the arterial baroreceptor input. The interpretation of the data from this particular experiment is somewhat limited because of the potential direct effects of sodium nitroprusside (used to induce acute hypotension) on the cerebral circulation, and the fact that the afferent innervation of the kidney remained intact. However, the reported effects of renal mechanosensory nerve stimulation, which have been shown to decrease sympathetic activity and increase urinary sodium excretion in rats (Kopp, 2015), argue against a major role for renal afferents in triggering the sympathoexcitatory responses to acute arterial hypotension. While the data obtained in the experiments of Study 3 are not fully conclusive on their own, they support the main hypothesis of the study when considered alongside the broader evidence. The intracranial baroreceptor mechanism monitors changes in CPP, which is largely determined by the difference between MAP and ICP (CPP = MAP – ICP). Our findings show that sympathetic activity increases in response to two experimental manipulations that decrease CPP – an increase in ICP or a decrease in MAP – and that these sympathetic responses occur independently of the arterial baroreceptor inputs.

Our study also examined the interactions between the arterial and intracranial baroreflexes by recording blood pressure and HR responses to electrical stimulation of the ADN at normal (basal) ICP and under conditions of increased ICP (by 10 and 15 mmHg). Figure 5*C* illustrates the input (frequency of ADN stimulation)–output ($\Delta$ABP, $\Delta$HR) relationships for the baroreceptor reflex at different levels of ICP. This analysis showed that activation of the intracranial baroreceptor mechanism reduces arterial baroreflex sensitivity (as lower-frequency aortic nerve stimulations triggered smaller decreases in MAP at raised ICP) and shifts its operating range to higher values of ABP. This cannot be explained by the adaptation/resetting of arterial baroreceptor neurons, as these experiments involved direct electrical stimulation of the ADN.

Recordings of renal nerve activity further showed that beat-to-beat regulation of sympathetic discharge by the arterial baroreflex was largely preserved at raised ICP despite significant increases in mean ABP. The physiological significance of baroreflex resetting is to maintain baroreceptor control of ABP at a higher level, which is necessary to counteract reduced brain perfusion under conditions of increased ICP. It is important to note that ICP varies with head position, body movement and changes in posture. Therefore, the intracranial baroreceptor mechanism would be expected to operate

in concert with arterial baroreceptors to ensure effective regulation of systemic and cerebral circulation across various physiological conditions and behaviours.

There is evidence that brainstem glial cells, specifically astrocytes, are likely candidates for the role of intracranial baroreceptors (Marina et al., 2020; Turovsky et al., 2020). Astrocytes are mechanosensitive (Bowser & Khakh, 2007; Turovsky et al., 2020; Yu et al., 2022) and have extensive end-feet that enwrap all penetrating and intraparenchymal cerebral blood vessels (Iadecola & Nedergaard, 2007). This anatomical organization makes them ideally positioned to sense changes in vascular lumen diameter and/or vascular wall stress associated with changes in cerebrovascular flow (Kim et al., 2015). Astrocytes in various brain regions, including the cerebral cortex and brainstem, have been reported to respond to acute increases in ICP with increased $Ca^{2+}$ signalling (Marina et al., 2020).

Regarding the location of the intracranial baroreceptor, Hoff & Reis (1970) first reported evidence suggesting that the receptive areas lie within the lower brainstem. In their experiments conducted in anaesthetized cats, robust pressor responses were evoked by direct mechanical stimulation applied to the floor of the fourth ventricle. Subsequently, the same group demonstrated that pressor responses could be evoked by mechanical distortion of brain tissue within a restricted area of the rostral medulla and caudal pons (Doba & Reis, 1972). More recently, it was reported that increases in ABP could be triggered by selective magnetomechanical stimulation of astrocytes within the RVLM region (Yu et al., 2022). Furthermore, the blockade of $Ca^{2+}$-dependent signalling mechanisms in RVLM astrocytes was shown to prevent cardiovascular and sympathetic responses to increased ICP (Marina et al., 2020), suggesting that the intracranial baroreceptor mechanism operates in the RVLM and involves interactions between glial cells and brainstem sympathetic control circuits.

The hypothesis that intracranial baroreceptors are glial cells is further supported by the observation that cardiovascular responses to increases in ICP often outlast the duration of the stimulus, particularly following larger increases in ICP (Figs 1*D*, 2*A* and 3*A*), as also reported in earlier publications (Hoff & Reis, 1970; Marina et al., 2020). In astrocytes, $Ca^{2+}$ responses to mechanosensory stimuli propagate to neighbouring cells and significantly outlast the periods of stimulation (see Fig. 2 in Bowser & Khakh (2007); Fig. 1 in Yu et al. (2022)). Lasting cardiovascular and sympathetic responses to raised ICP observed in experiments of this type are, therefore, consistent with the mechanosensory properties of glial cells.

Since astrocytes are not excitable cells and do not generate action potentials, the increases in sympathetic nerve activity are hypothesized to be driven by the mechanosensory release of astroglial signalling molecules, such as ATP and lactate, which have been shown to increase the firing of brainstem pre-sympathetic neurons (Marina et al., 2015; Sun et al., 1992). A comprehensive understanding of the intracranial baroreflex mechanism will require further experimental studies involving recordings of individual, functionally identified pre-sympathetic RVLM neurons and characterization of their responses to changes in brain perfusion. This experimental model could then be used to identify astroglial transmitter(s) responsible for sympathoexcitation under these conditions.

In conclusion, the results of this study provide further characterization of the intracranial baroreceptor mechanism. This mechanism is highly sensitive to and monitors changes in cerebral perfusion and appears to play a critical role in regulating sympathetic nerve activity and ABP. When activated, the intracranial baroreceptor resets the arterial baroreflex centrally to regulate systemic blood pressure at a higher level required to maintain blood flow to the brain.

We hypothesize that the intracranial baroreceptor mechanism contributes to the maintenance of cerebral blood flow during changes in head and body posture, as well as during body movements. This mechanism is likely to be more sensitive in humans and certain animals, such as giraffes (Paton et al., 2009), since an upright posture increases susceptibility to the effects of gravity and, consequently, to substantial fluctuations in ICP and cerebral perfusion. We propose that the intracranial baroreceptor mechanism may be particularly significant for maintaining adequate brain perfusion in ageing, which is associated with a progressive reduction of cerebral blood flow (by ∼5% per decade of life from middle age; Christie et al., 2022), and especially when cerebral vessels become stenotic or atherosclerotic in disease (Warnert et al., 2016), contributing to the development of systemic arterial hypertension.

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

## Additional information

### Data availability statement

The data that support the findings of this study are available from the corresponding authors upon reasonable request.

### Competing interests

The authors declare they have no competing interests.

## Author contributions

A.V.G. conceived and directed the project; A.V.G., F.D.M., J.F.R.P. and N.M. designed research; P.W., F.D.M., A.K., K.L.R., N.M. and A.V.G. performed research; P.W., F.D.M., A.K. and K.L.R. analysed data. All authors contributed to writing the paper and revised the article critically for important intellectual content. All authors have read and approved the final version of this manuscript and agree to be accountable for all aspects of the work in ensuring that questions related to the accuracy or integrity of any part of the work are appropriately investigated and resolved. All persons designated as authors qualify for authorship, and all those who qualify for authorship are listed.

## Funding

This work was supported by a British Heart Foundation PhD studentship (ref. FS/PhD/22/29299) and a Health Research Council of New Zealand Project Grant (ref. 23-165).

## Acknowledgements

J.F.R.P. is supported by the Sidney Taylor Trust and is an inaugural Partridge Family research laureate.

## Keywords

arterial pressure, arterial baroreflex, baroreceptor, cerebral blood flow, intracranial pressure, sympathetic activity

## Supporting information

Additional supporting information can be found online in the Supporting Information section at the end of the HTML view of the article. Supporting information files available:

**Peer Review History**

