## [Peer Review History · The Journal of Physiology]

On the regulation of arterial blood pressure by an intracranial baroreceptor mechanism

Philippa Wittenberg, Fiona D McBryde, Alla Korsak, Karla L Rodrigues, Julian F. R. Paton, Nephtali Marina, and Alexander V Gourine

DOI: 10.1113/JP285082

Corresponding author(s): Alexander Gourine (a.gourine@ucl.ac.uk)

Review Timeline:

Submission Date:	01-Nov-2024
Editorial Decision:	25-Nov-2024
Revision Received:	14-Jan-2025
Accepted:	20-Jan-2025

Senior Editor: Kim Barrett

Reviewing Editor: Philip Ainslie

Transaction Report:

Dear Dr Gourine,

Re: JP-RP-2024-285082 "On the regulation of arterial blood pressure by an intracranial baroreceptor mechanism" by Philippa Wittenberg, Fiona D McBryde, Alla Korsak, Karla L Rodrigues, Julian F. R. Paton, Nephtali Marina, and Alexander V Gourine

Thank you for submitting your manuscript to The Journal of Physiology. It has been assessed by a Reviewing Editor and by 2 expert referees and we are pleased to tell you that it is acceptable for publication following satisfactory revision.

REVISION CHECKLIST:

We look forward to receiving your revised submission.

Yours sincerely,

Kim Barrett
Senior Editor
The Journal of Physiology

REQUIRED ITEMS FOR REVISION

- Author photo and profile. First or joint first authors are asked to provide a short biography (no more than 100 words for one author or 150 words in total for joint first authors) and a portrait photograph. These should be uploaded and clearly labelled together in a Word document with the revised version of the manuscript. See Information for Authors for further details.

- Your manuscript must include a complete Additional Information section, including competing interests; funding; author contributions and acknowledgements.

- Please upload separate high-quality figure files via the submission form.

- Papers must comply with the Statistics Policy: https://jp.msubmit.net/cgi-bin/main.plex?form_type=display_requirements#statistics.

In summary:

- If n {less than or equal to} 30, all data points must be plotted in the figure in a way that reveals their range and distribution. A bar graph with data points overlaid, a box and whisker plot or a violin plot (preferably with data points included) are acceptable formats.

- If $n > 30$, then the entire raw dataset must be made available either as supporting information, or hosted on a not-for-profit repository, e.g. FigShare, with access details provided in the manuscript.

- 'n' clearly defined (e.g. x cells from y slices in z animals) in the Methods. Authors should be mindful of pseudoreplication.

- All relevant 'n' values must be clearly stated in the main text, figures and tables.

- The most appropriate summary statistic (e.g. mean or median and standard deviation) must be used. Standard Error of the Mean (SEM) alone is not permitted.

- Exact p values must be stated. Authors must not use 'greater than' or 'less than'. Exact p values must be stated to three significant figures even when 'no statistical significance' is claimed.

- Please include an Abstract Figure file, as well as the Figure Legend text within the main article file. The Abstract Figure is a piece of artwork designed to give readers an immediate understanding of the research and should summarise the main conclusions. If possible, the image should be easily 'readable' from left to right or top to bottom. It should show the physiological relevance of the manuscript so readers can assess the importance and content of its findings. Abstract Figures should not merely recapitulate other figures in the manuscript. Please try to keep the diagram as simple as possible and without superfluous information that may distract from the main conclusion(s). Abstract Figures must be provided by authors no later than the revised manuscript stage and should be uploaded as a separate file during online submission labelled as

File Type 'Abstract Figure'. Please also ensure that you include the figure legend in the main article file. All Abstract Figures should be created using BioRender. Authors should use The Journal's premium BioRender account to export high-resolution images. Details on how to use and access the premium account are included as part of this email.

EDITOR COMMENTS

Reviewing Editor:

Comments for Authors to ensure the paper complies with the Statistics Policy:
Please use SD and not SEM per journal policy

Comments to the Author:

Thank you for submitting your impressive work. As you will see, both reviewers were very positive in their reviews. Please address each of their important comments. In addition to a brief discussion on the clinical implications of this work, it would also be beneficial to review the related evidence in humans of the potential role of the postulated intracranial baroreceptors.

Senior Editor:

Comments for Authors to ensure the paper complies with the Statistics Policy:
See editor comments

REFEREE COMMENTS

Referee #1:

The studies presented in the submitted manuscript by Wittenberg and colleagues explore cardiovascular and peripheral sympathetic responses to changes in intracranial pressure (ICP), spanning mechanisms, experimental models, and potential clinical implications across five separate, insightful, and well conducted animal studies.. While the research addresses a relevant and important topic, there are several areas that require clarification, additional details, and careful interpretation of the findings. Below are specific comments and suggestions for the specific studies and general structure of the manuscript in no particular order.

Figure 1d: Following the reduction of ICP, there appears to be a lingering effect on arterial blood pressure (ABP) and heart rate (HR). What is the proposed mechanism for this persistence? Further explanation or discussion would strengthen this section.

Study 3: Does SNP have a direct effect on cerebral blood volume and pressure? The data are convincing that both increases and decreases in mean arterial pressure can signal for increases in peripheral SNA. It seems that cerebral perfusion pressure, not mean arterial blood pressure per se is the driving force for intracranial baroreflex activation. More emphasis on this throughout the manuscript should be considered.

General question: Is there any evidence for resting (i.e., tonic) levels of intracranial baroflex activity? Or is this reflex activated only in response to acute changes in CPP?

Study 4: There is a notable alteration in blood pressure (BP), yet ICP measurements are absent. Clarification on why ICP was not measured would be useful, as it appears integral to the conclusions drawn in this study.

Study 5: Was the observed increase in BP statistically significant in response to ICP elevation? This is not clearly stated in the text, figure legend, or figure itself. A p-value should be provided to confirm the robustness of the findings.

The manuscript could benefit from a more comprehensive discussion of the clinical implications, particularly the role of

astrocytes and interactions with other pathways in modulating ICP responses. Is the intracranial baroreflex thought to be located throughout the cerebrovasculature? Or more prominent in specific locations (e.g., jugular sinus)

Could the authors provide some data regarding the latency period between changes in ICP and the respective changes in blood pressure, heart rate, and sympathetic nerve activity. Do changes in ICP precede the downstream cardiovascular effects in every experiment? Knowing the time between the initial rise (or peak plateau in ICP) vs each individual parameter would be very interesting and insightful into this mechanism.

Baroreflex resetting, observed in simultaneous increases in HR and BP, should be discussed in terms of its physiological purpose. ICP is typically constant, so what does this adaptation achieve in acute or chronic settings?

The authors should consider other baroreceptor types, such as great vein or pulmonary baroreceptors, may enhance the mechanistic understanding of the responses to ICP changes. How many these other baroreceptors integrate into this ICP mechanism?

The manuscript should address potential differences in ICP and cardiovascular responses across species, including the translational relevance of anesthetized models versus conscious settings.

Long-term steady-state ICP effects are not discussed but are crucial for understanding chronic adaptations (e.g., in chronic idiopathic intracranial hypertension, or this would even be highly useful in the context of space flight).

Line 102: Typographical error ("Collectively") should be corrected.

Results Line 324-325: Provide the p-value for the reported non-significant HR change to enable proper interpretation of these findings.

Results Line 346-347: The p-values for reported increases in RSNA are missing and should be included.

The authors should consider the effects of altering ICP in hyperoxic conditions (Lines 469-475). This could provide valuable insights into how oxygenation status influences cardiovascular and sympathetic responses.

Referee #2:

This is an excellent study that uncovers further details of the underlying physiology of the postulated intracranial baroreceptor. Described in humans as an increase in blood pressure during increases in intracerebral pressure, the authors here use an animal model to interrogate the mechanisms, providing evidence in an elegant series of experiments to support the idea that the intracranial baroreceptor operates independently of the arterial baroreceptors to adjust the total intracerebral pressure and hence perfusion of the brain. I have little to offer that would improve the manuscript, other than in a few areas.

Line 85: "with projections in the aortic and carotid walls" Strictly, the mechanoreceptors are in the aortic arch and carotid sinus and the projections travel in the vagus and glossopharyngeal nerves, respectively. Please revise accordingly.

Line 102: Replace "Collectivley" with "Collectively"

Line 253: "blockade of action potential propagation..."

Line 329: "Experimentally-induced"

Lines 499-500: Replace "inhibitory synaptic connection responsible for the inhibition..." with "inhibitory synaptic connection between the caudal ventrolateral medulla (CVLM) and the rostral ventrolateral medulla (RVLM), responsible for the inhibition..."

Line 520: Replace "leading to sympathetic disinhibition and subsequent increases..." with "leading to sympathetic disinhibition, i.e. an increase in vasoconstrictor drive to resistance vessels, and subsequent increases..."

Line 525-526: "That the afferent innervation of the kidney remained intact in this particular experiment limits interpretation of these data" With one renal nerve cut for recording of RSNA, and the other blocked with lidocaine, why do you state there is ongoing afferent activity?

Lines 542-543: "inputs from arterial baroreceptors are not fully inhibited" It would be worth pointing out that the beat-to-beat regulation of sympathetic outflow by the arterial baroreflex was preserved despite the tonic increases in MBP with increases in ICP

Figure 2 is a little confusing: it may be better to have labels above panels b-e to indicate that "initial dip" in panels b and d and "overshoot" in panels c and e.

Please note that Journal policy is to report SD, not SEM, so please change throughout.

END OF COMMENTS

Editor comments:

Thank you for submitting your impressive work. As you will see, both reviewers were very positive in their reviews. Please address each of their important comments. In addition to a brief discussion on the clinical implications of this work, it would also be beneficial to review the related evidence in humans of the potential role of the postulated intracranial baroreceptors.

Response: We would like to thank the Reviewers and the Editors of *The Journal of Physiology* for their time taken to evaluate our submission and for the overall positive assessment of our work. We are grateful for the detailed and constructive comments provided and are delighted to have the opportunity to submit a revised manuscript. We have revised the text and figures in accordance with the comments provided by the reviewers and the Editors. As requested, we have included citations to the relevant evidence in humans, highlighting studies that demonstrated increases in arterial blood pressure and sympathetic activity in response to experimental increases in intracranial pressure (ICP) (PMID: 16463859; PMID: 29472865), as well as studies that reported decreases in arterial blood pressure following decompressive craniectomy in patients with traumatic brain injury or other conditions (PMID: 18173312; PMID: 31954889; PMID: 34749485). Please review our full response to all the comments provided by the Reviewers.

Referee #1:

The studies presented in the submitted manuscript by Wittenberg and colleagues explore cardiovascular and peripheral sympathetic responses to changes in intracranial pressure (ICP), spanning mechanisms, experimental models, and potential clinical implications across five separate, insightful, and well conducted animal studies. While the research addresses a relevant and important topic, there are several areas that require clarification, additional details, and careful interpretation of the findings. Below are specific comments and suggestions for the specific studies and general structure of the manuscript in no particular order.

Response: We thank the Referee for taking the time to review our manuscript and for their overall positive assessment of our work. We are grateful for the constructive comments provided and have revised the text of the manuscript accordingly.

Figure 1d: Following the reduction of ICP, there appears to be a lingering effect on arterial blood pressure (ABP) and heart rate (HR). What is the proposed mechanism for this persistence? Further explanation or discussion would strengthen this section.

Response: We thank the Reviewer for raising this comment. Indeed, in our experiments, cardiovascular responses to increases in ICP often outlasted the duration of the stimulus, especially following larger increases in ICP. These lasting effects on arterial blood pressure have been observed in earlier studies (PMID: 5456720) and also reported in our more recent publication (PMID: 31919423). While the underlying mechanisms are not fully understood, these lasting responses are consistent with the hypothesis that brain glial cells (astrocytes) act as intracranial baroreceptors. In these glial cells the Ca²⁺ responses to mechanosensory stimuli significantly outlast the periods of stimulation (see Figure 2 in PMID: 17504911; Figure 1 in PMID: 34927381). Lasting cardiovascular and sympathetic responses to increased ICP are, therefore, consistent with the mechanosensory properties of astrocytes

and their proposed role as intracranial baroreceptors. We have now included a brief discussion of this point in the revised manuscript.

Study 3: Does SNP have a direct effect on cerebral blood volume and pressure? The data are convincing that both increases and decreases in mean arterial pressure can signal for increases in peripheral SNA. It seems that cerebral perfusion pressure, not mean arterial blood pressure per se is the driving force for intracranial baroreflex activation. More emphasis on this throughout the manuscript should be considered.

Response: We thank the Reviewer for raising this important point. SNP may indeed have a direct effect on cerebral circulation, and this potential effect should be considered when interpreting the data obtained in the experiments using SNP to lower systemic arterial blood pressure. While the data from these experiments are not fully conclusive on their own, they support the main hypothesis of the study when considered alongside the rest of the evidence. Indeed, we hypothesize that the intracranial baroreceptor mechanism is sensitive to changes in cerebral perfusion pressure (CPP), which is determined by the difference between mean arterial blood pressure (MAP) and intracranial pressure (ICP): $CPP = MAP - ICP$. Our study shows that sympathetic activity increases in response to two experimental manipulations that decrease cerebral perfusion pressure: an increase in ICP or a decrease in MAP. Importantly, these sympathetic responses occur independently of the arterial baroreceptor inputs. We now revised the Discussion section of the paper to address this comment of the Reviewer.

General question: Is there any evidence for resting (i.e., tonic) levels of intracranial baroreflex activity? Or is this reflex activated only in response to acute changes in CPP?

Response: Yes, we report data suggesting that the intracranial baroreceptor mechanism contributes to the maintenance of systemic arterial pressure under resting conditions. Please review Figure 1b,c which shows that in urethane-anaesthetised rats, an acute reduction in ICP to 0 mmHg (from its basal level of ~6 mmHg) following craniotomy led to a decrease in mean arterial pressure by ~10 mmHg. Restoration of basal ICP following the implantation of intraventricular cannulae and subsequent repair, increased arterial blood pressure back to the baseline level recorded before craniotomy.

We (JFRP, FDM) have long argued that decreases in cerebral perfusion due to aging and/or disease contribute to the development of arterial hypertension. In our manuscript, we further discuss that the intracranial baroreceptor mechanism may play a particularly significant role in maintaining adequate brain perfusion during aging, which is associated with a progressive reduction in cerebral blood flow (by ~5% every decade of life from middle age [PMID: 36113055]), and especially when cerebral vessels become stenotic or atherosclerotic due to disease (PMID: 27672161), potentially contributing to the development of systemic arterial hypertension.

Study 4: There is a notable alteration in blood pressure (BP), yet ICP measurements are absent. Clarification on why ICP was not measured would be useful, as it appears integral to the conclusions drawn in this study.

Response: We thank the Reviewer for raising this comment and apologize for the lack of clarity in our description of this experiment. ICP was measured in all the experiments, as our aim was to apply precise stimuli (increases in ICP by 10 and 15 mmHg above baseline) to determine how baroreflex is affected under conditions of intracranial baroreceptor activation. We have revised Figure 5 to illustrate changes in ICP observed in response to aortic depressor nerve stimulation under basal conditions and conditions of increased ICP.

Study 5: Was the observed increase in BP statistically significant in response to ICP elevation? This is not clearly stated in the text, figure legend, or figure itself. A p-value should be provided to confirm the robustness of the findings.

Response: Thank you for raising this point, and we apologize for the oversight. Yes, increases in ICP trigger highly significant proportional increases in arterial blood pressure in conscious rats. We now provide p values in the revised text and revised Figure 6.

The manuscript could benefit from a more comprehensive discussion of the clinical implications, particularly the role of astrocytes and interactions with other pathways in modulating ICP responses. Is the intracranial baroreflex thought to be located throughout the cerebrovasculature? Or more prominent in specific locations (e.g., jugular sinus)

Response: We thank the Reviewer for raising this comment. As discussed above, we (JFRP, FDM) have long argued that decreases in cerebral perfusion due to aging and/or cerebrovascular disease contribute to the development of arterial hypertension. In our paper, we further discuss that the intracranial baroreceptor mechanism may play a particularly significant role in maintaining adequate brain perfusion during aging, which is associated with a progressive reduction in cerebral blood flow (by ~5% every decade of life from middle age [PMID: 36113055]), and especially when cerebral vessels become stenotic or atherosclerotic due to disease (PMID: 27672161), contributing to the development of systemic arterial hypertension.

Regarding the location of the intracranial baroreceptor, Hoff and Reis (PMID: 5456720) first reported evidence suggesting that the receptive areas lie within the lower brainstem. In their experiments conducted in anaesthetised cats, robust pressor responses were evoked by direct mechanical stimulation applied to the floor of the fourth ventricle. Subsequently, the same group demonstrated that increases in arterial blood pressure could be evoked by mechanical brain tissue distortion of a restricted area within the rostral medulla and caudal pons (PMID: 4642574). More recently, our group reported that blockade of astroglial signalling in the RVLM region prevented sympathetic and blood pressure responses induced by increases in ICP (PMID: 31919423), supporting the hypothesis that the intracranial baroreceptor mechanism involves interactions between glial cells and brainstem sympathetic circuits. The molecular and signalling mechanisms underlying these interactions remain unknown and are currently under investigation. As suggested by the Reviewer, we have now included this discussion in the revised manuscript.

Could the authors provide some data regarding the latency period between changes in ICP and the respective changes in blood pressure, heart rate, and sympathetic nerve activity. Do changes in ICP precede the downstream cardiovascular effects in every experiment? Knowing the time between the initial rise (or peak plateau in ICP) vs each individual parameter would be very interesting and insightful into this mechanism.

Response: Thank you for this comment. In our earlier publication we reported that cardiovascular and sympathetic responses followed increases in ICP with a mean delay of ~30s (PMID: 31919423). In all our experiments, increases in ICP always preceded the cardiovascular responses. As suggested by the Reviewer, we have now analysed the latencies between increases in ICP and the resulting cardiovascular responses recorded in this study. The data are now provided in the revised manuscript: "*The blood pressure responses followed increases in ICP with a mean delay of 22±5 s (n=8), consistent with the previously reported data (Marina et al., 2020)*".

Baroreflex resetting, observed in simultaneous increases in HR and BP, should be discussed in terms of its physiological purpose. ICP is typically constant, so what does this adaptation achieve in acute or chronic settings?

Response: We thank the Reviewer for raising this discussion point. The data presented in this manuscript (particularly the analysis of renal nerve activity at resting ICP and under conditions of increased ICP, which showed strong inhibition of discharge during systole), suggest that inputs from arterial baroreceptors are not inhibited or overridden but that the baroreflex is reset. We propose that the physiological significance of the baroreflex resetting at raised ICP is to maintain baroreflex control of arterial blood pressure at a higher level, which is necessary to counteract reduced brain perfusion. It is important to note that ICP varies with head position, body movement, and changes in posture. Therefore, the intracranial baroreceptors operate in concert with arterial baroreceptors to ensure effective regulation of systemic and cerebral circulation across various conditions. We have now included a brief discussion of this point in the revised manuscript.

The authors should consider other baroreceptor types, such as great vein or pulmonary baroreceptors, may enhance the mechanistic understanding of the responses to ICP changes. How many of these other baroreceptors integrate into this ICP mechanism?

Response: We thank the Reviewer for this suggestion. We agree that the interactions between the intracranial baroreceptors and other baroreceptor inputs mentioned by the Referee, merit further detailed investigation. But please note that in this paper we present data and characterize the cardiovascular responses induced by increased ICP under conditions where these peripheral inputs were interrupted; specifically after (1) unilateral arterial baroreceptor and cardiopulmonary deafferentation, achieved by transecting the left vagus nerve, left carotid sinus nerve and left aortic depressor nerve, and (2) subsequent complete bilateral baroreceptor denervation and vagotomy, achieved by blocking action potential propagation in the vagus, carotid sinus and aortic nerves on the contralateral side using lidocaine. Under conditions of bilateral baroreceptor/ cardiopulmonary deafferentation the blood pressure responses to raised ICP were increased, suggesting that baroreceptor inputs restrain the responses induced by activation of the intracranial baroreceptors. Future studies will be designed to isolate other baroreceptive sites and investigate their interactions with the intracranial baroreceptor mechanism.

The manuscript should address potential differences in ICP and cardiovascular responses across species, including the translational relevance of anesthetized models versus conscious settings.

Response: The cardiovascular and sympathetic responses recorded in our experiments are consistent with data reported in published studies conducted in mice (PMID:29472865), rats (PMID:31919423), sheep (PMID:30207755; PMID:33712655) and humans (PMID:16463859; PMID:29472865), all of which involved experimental manipulations of ICP within the physiological range. In our paper, we present data obtained in *both* conscious and anesthetized rats. In urethane-anaesthetised rats, we recorded a resting (basal) level of ICP of ~6 mmHg, which matches exactly the ICP recorded in conscious rats using telemetry (PMID:26159754) and falls within the normal range of ICP for humans. A linear relationship between mean arterial blood pressure and ICP within the range of 0-30 mmHg was observed under both conditions (revised Figures 1f and 6b), an observation supported by the results of previous studies conducted in sheep and humans (PMID:30207755; PMID:29472865).

Long-term steady-state ICP effects are not discussed but are crucial for understanding chronic adaptations (e.g., in chronic idiopathic intracranial hypertension, or this would even be highly useful in the context of space flight).

Response: We thank the Reviewer for raising this discussion point. However, in our opinion, discussing the potential effects of chronic idiopathic intracranial hypertension and spaceflight conditions on the operation and adaptation of the intracranial baroreceptor mechanism would be largely speculative. There is strong evidence that blood pressure decreases by 8–10 mmHg during spaceflight (PMID:25774397). However, the effects of microgravity on ICP remain less clear. A recent study reported reductions in non-invasively estimated ICP after spaceflight (PMID: 33103234) and, therefore, these observations would be consistent with the proposed role of the intracranial baroreceptor mechanism. Nonetheless, we consider that addressing these points would extend beyond the scope of this manuscript's focused discussion of the data obtained in the study.

Line 102: Typographical error ("Collectively") should be corrected.

Thank you; now corrected.

Results Line 324-325: Provide the p-value for the reported non-significant HR change to enable proper interpretation of these findings.

Thank you, now provided in the revised manuscript.

Results Line 346-347: The p-values for reported increases in RSNA are missing and should be included.

Thank you, now provided in the revised manuscript.

The authors should consider the effects of altering ICP in hyperoxic conditions (Lines 469-475). This could provide valuable insights into how oxygenation status influences cardiovascular and sympathetic responses.

Response: We thank the Reviewer for this comment. Oxygen measurements in the RVLM showed that increases in ICP (by up to 20 mmHg) result in very small (~1.5 mmHg) and transient decreases in PtO₂ (revised Figure 2), indicating that changes in ICP within the physiological range, as applied in the experiments of this study, are not associated with brain tissue hypoxia. Therefore, the observed sympathetic and cardiovascular responses are driven by stimuli other than changes in oxygen delivery to the brainstem.

Referee #2:

This is an excellent study that uncovers further details of the underlying physiology of the postulated intracranial baroreceptor. Described in humans as an increase in blood pressure during increases in intracerebral pressure, the authors here use an animal model to interrogate the mechanisms, providing evidence in an elegant series of experiments to support the idea that the intracranial baroreceptor operates independently of the arterial baroreceptors to adjust the total intracerebral pressure and hence perfusion of the brain. I have little to offer that would improve the manuscript, other than in a few areas.

Response: We thank the Reviewer for taking the time to evaluate our manuscript and for their very positive assessment of our work. We greatly appreciate the constructive comments provided and have revised the text of the manuscript accordingly.

Line 85: "with projections in the aortic and carotid walls" Strictly, the mechanoreceptors are in the aortic arch and carotid sinus and the projections travel in the vagus and glossopharyngeal nerves, respectively. Please revise accordingly.

Response: Thank you. We have revised the text to read: *"In response to increases in arterial blood pressure, these stretch-sensitive baroreceptor neurons (Zeng et al., 2018), with peripheral axons innervating the walls of the aortic arch and carotid sinus, initiate the arterial baroreflex, leading to reductions in heart rate, contractility, and peripheral vascular resistance"*.

Line 102: Replace "Collectivley" with "Collectively"

Thank you; now corrected.

Line 253: "blockade of action potential propagation..."

Response: Thank you. We have revised the text to read: *"After a recovery period of up to 1 h to allow ABP and HR to return to baseline, complete peripheral baroreceptor denervation was achieved by applying lidocaine (1%) to the vagus nerve, carotid sinus nerve and aortic depressor nerve on the right side, followed by the assessment of the cardiovascular response induced by increased ICP (10 mmHg)"*.

Line 329: "Experimentally-induced"

Thank you; now corrected.

Lines 499-500: Replace "inhibitory synaptic connection responsible for the inhibition..." with "inhibitory synaptic connection between the caudal ventrolateral medulla (CVLM) and the rostral ventrolateral medulla (RVLM), responsible for the inhibition..."

Response: Thank you. We have revised the text to read: *"The central nervous pathway of the arterial baroreflex includes an inhibitory synaptic connection between neurons of the caudal ventrolateral medulla (which receive excitatory inputs from arterial baroreceptors via the nucleus of the solitary tract) and sympathoexcitatory neurons of the rostral ventrolateral medulla. This neural pathway is ultimately responsible for inhibiting vasomotor and cardiac sympathetic activities in response to acute increases in arterial baroreceptor afferent discharge."*

Line 520: Replace "leading to sympathetic disinhibition and subsequent increases..." with "leading to sympathetic disinhibition, i.e. an increase in vasoconstrictor drive to resistance vessels, and subsequent increases..."

Response: Thank you. We have revised the text to read: *"It is often assumed that sympathetic responses to acute decreases in arterial blood pressure are mediated through the same peripheral baroreceptor reflex pathway, where arterial baroreceptor 'unloading' leads to disinhibition of sympathetic activity, resulting in increased cardiac output and peripheral vascular resistance"*.

Line 525-526: "That the afferent innervation of the kidney remained intact in this particular experiment limits interpretation of these data" With one renal nerve cut for recording of RSNA, and the other blocked with lidocaine, why do you state there is ongoing afferent activity?

Response: We thank the Reviewer for raising this comment and apologize for the lack of clarity in our description. In this experiment, no lidocaine blockade was applied to the renal nerves; therefore, afferent inputs from the kidneys were not interrupted. We have now revised the text to read: *"Complete arterial baroreceptor and cardiopulmonary deafferentation was produced as described above; but innervation of the kidneys was left intact in this experiment"*.

Lines 542-543: "inputs from arterial baroreceptors are not fully inhibited" It would be worth pointing out that the beat-to-beat regulation of sympathetic outflow by the arterial baroreflex was preserved despite the tonic increases in MBP with increases in ICP

Response: Thank you for this suggestion. We have added the following sentence to the Discussion section of the revised manuscript: *"Recordings of renal nerve activity further showed that beat-to-beat regulation of sympathetic outflow by the arterial baroreflex was largely preserved despite significant increases in mean arterial blood pressure."*

Figure 2 is a little confusing: it may be better to have labels above panels b-e to indicate that "initial dip" in panels b and d and "overshoot" in panels c and e.

Thank you for the suggestion; Figure 2 has been revised accordingly.

Please note that Journal policy is to report SD, not SEM, so please change throughout.

Thank you; the text and figures have been revised to report averages/summary data as means \pm SD.

Dear Professor Gourine,

Re: JP-RP-2025-285082R1 "On the regulation of arterial blood pressure by an intracranial baroreceptor mechanism" by Philippa Wittenberg, Fiona D McBryde, Alla Korsak, Karla L Rodrigues, Julian F. R. Paton, Nephtali Marina, and Alexander V Gourine

We are pleased to tell you that your paper has been accepted for publication in The Journal of Physiology.

Yours sincerely,

Kim Barrett
Senior Editor
The Journal of Physiology

If you would like to receive our 'Research Roundup', a monthly newsletter highlighting the cutting-edge research published in The Physiological Society's family of journals (The Journal of Physiology, Experimental Physiology, Physiological Reports, The Journal of Nutritional Physiology and The Journal of Precision Medicine: Health and Disease), please click this link, fill in your name and email address and select 'Research Roundup':
<https://www.physoc.org/journals-and-media/membernews>

- You can help your research get the attention it deserves! Check out Wiley's free Promotion Guide for best-practice recommendations for promoting your work at: www.wileyauthors.com/eeo/guide. You can learn more about Wiley Editing Services which offers professional video, design, and writing services to create shareable video abstracts, infographics, conference posters, lay summaries, and research news stories for your research at: www.wileyauthors.com/eeo/promotion.

EDITOR COMMENTS

Reviewing Editor:

Thank you for addressing the initial comments from the reviewers. This is an elegant set of studies that uncovers the underlying physiology behind the postulated intracranial baroreceptor and forms a major contribution to this field.

REFEREE COMMENTS

Referee #1:

Thank you for addressing all of my concerns. Congrats on a very comprehensive and insightful series of studies.

Referee #2:

Thank you for addressing my comments. I have nothing further to add and commend the authors on a remarkable set of studies.